# Current and future prediction of inter-provincial transport of ambient PM<sub>2.5</sub> in China

Shansi Wang<sup>1</sup>, Siwei Li<sup>1,2</sup>, Jia Xing<sup>3</sup>, Yu Ding<sup>1</sup>, Senlin Hu<sup>1</sup>, Shuchang Liu<sup>3</sup>, Yu Qin<sup>4</sup>, Zhaoxin Dong<sup>3</sup>,

5 Jiaxin Dong<sup>1</sup>, Ge Song<sup>1</sup>, Lechao Dong<sup>1</sup>

1 School of Remote Sensing and Information Engineering, Wuhan University, Wuhan, 430079, China

2 State Key Laboratory of Information Engineering in Surveying, Mapping and Remote Sensing, Wuhan University, Wuhan, 430079, China

10 3 School of Environment, Tsinghua University, Beijing, 100084, China

4 Map Institute of Guangdong Province, Guangzhou, 510075, China

Correspondence to: Siwei Li

15 School of Remote Sensing and Information Engineering, Wuhan University

Wuhan, 430079, China

Email: siwei.li@whu.edu.cn

- 20 Abstract: Regional transport is as much important as local sources that contributing to PM<sub>2.5</sub> pollution and causing associated environmental inequality. In the context of future climate change, the effect of the responses of regional transport to the warming meteorology has not been thoroughly investigated. Here we establish cross-province PM<sub>2.5</sub> source-receptor matrix in China in 2015 and two climate pathways in 2050s (SSP585 and SSP126), using Community Multi-scale Air Quality model embedded with the Integrated Source Apportionment Method. Results suggest that across-regional transport contributes 27 % 56.8 % of PM<sub>2.5</sub> in five severely polluted regions, which is even more important compared to inner transport within the target region
- (13.2 % 20.9 %), especially in Chuanyu and Fenwei regions where suffers large PM<sub>2.5</sub> transport (over 50 %) from outside regions. Such results imply that joint-control policy should not only focus on neighboring provinces. Future warming scenario (SSP585) will exacerbate PM<sub>2.5</sub> pollution (2 5 µg/m<sup>3</sup>) and also enhance its regional transport (> 3 %) mostly by modulating the across-regional transport rather than inner regional transport. Such enhancement of regional transport of PM<sub>2.5</sub> can be significantly weaken (approximately by half) under SSP126 pathway, demonstrating the importance of climate change mitigation
- on weakening the regional transport of PM<sub>2.5</sub> to maximize the co-benefits in both air quality and climate.

Keywords: air quality, PM2.5, CMAQ-ISAM, climate change, inter-provincial transport

### **Graphical abstract**

# Relatively large PM<sub>2.5</sub> transport (µg/m<sup>3</sup>)

Number : 2015 annual averaged ( changed: 2050 SSP585 pathway )

**Short Summary:** Future warming meteorological conditions may enhance the influence of regional transport on PM<sub>2.5</sub> pollution. Our results prove that climate-friendly policy could lead to considerable co-benefits in mitigating the regional transport of PM<sub>2.5</sub> in future. Meanwhile, climate change will exert larger impacts on across-regional (long-distance) transport than inner (neighboring provinces) regional transport, highlighting the significance of multi-regional cooperation in the future.

### 40 1. Introduction

China has been suffering from severe PM<sub>2.5</sub> (fine particulars with a diameter of 2.5 micrometers or less) pollution over the past few years (Zhang et al., 2014; Wang et al., 2014; Wang et al., 2014; Wang et al., 2014). After enforcement of the Air Pollution Prevention and Control Action Plan in 2013 (Chinese State Council, 2013) and the three-year action plan to fight air pollution in 2017 (Ministry Of Ecology And Environment, 2017), air quality improved significantly (Ministry Of Ecology And Environment,

- 2017), while the PM<sub>2.5</sub> concentrations are still far above the new World Health Organization (WHO) Air Quality Guidelines (5 μg m<sup>-3</sup>). One big challenge to further reduce PM<sub>2.5</sub> is for its large contribution from external sources (Qin et al., 2015) (i.e., regional transport) which can be even more important under strengthened local emission controls as witnessed during the COVID-19 lockdown period (Zhao et al., 2020; Shen et al., 2021). Moreover, the problem of environmental inequality occurs due to air pollution transport, causing the receptor region to suffer additional health damage or economic burden from other
- regions. Zhang et al (2017) found that about 411,100 premature deaths (12 % of global 3.45 million deaths) were associated with air pollutants emitted from other parts of the world. Dedoussi et al (2020) indicated that in US about 41 to 53 percent of air quality-related premature mortality resulting from one state's emissions occurs outside that state. Jiang et al (2021) reported that people staying in rural areas in China are subjected to 22 % more impacts of regional transport to PM<sub>2.5</sub>-related deaths as compared with urban compatriots.
- The relatively long lifetime of PM<sub>2.5</sub> particles (approximately a week) allows them for long-range transport. Previous studies mainly concentrated on the transport within city clusters, such as Beijing-Tianjin-Hebei (BTH) (Chang et al., 2019; Dong et al., 2020), Yangtze River Delta (YRD) (Li et al., 2015; Li et al., 2016), and Pearl River Delta (PRD) region (Wu et al., 2013; Lu et al., 2019). However, the long-range transport among those city clusters has not been well studied. In fact, PM<sub>2.5</sub> pollution is not restricted to surrounding influence, which requires joint controls with multi-regional policy implementation across the whole
- country (Zhang et al., 2017; Han and Zhang, 2021). Therefore, quantification of both inner regional and long-range transport on PM<sub>2.5</sub> pollution across China is prerequisite for designing an effective multiple-regional joint control strategy.

Furthermore, climate change could play an important role in air quality and regional transport (Wang et al., 2014; Li et al., 2019; Dong et al., 2020) in the context of global warming, leading to increasing extreme weathers (Wang et al., 2017; Johnson et al., 2018; Ma et al., 2018; Zhang and Wang, 2019; Pinkerton et al., 2019) and weakening global atmospheric circulation (Wang et al.,

- 2017; Johnson et al., 2018; Ma et al., 2018; Zhang and Wang, 2019; Pinkerton et al., 2019). Most previous studies are based on current meteorological conditions (Wang et al., 2017; Johnson et al., 2018; Ma et al., 2018; Zhang and Wang, 2019; Pinkerton et al., 2019), without considering the effects of future climate change. It is still unclear whether the variation of atmospheric circulation driven by climate change could strengthen or weaken regional transport of PM<sub>2.5</sub>, thus modulating the source-receptor relationship. Meanwhile, an ambitious target named carbon peaking and carbon neutrality was put forward by the Chinese
- government, which aims to achieve short-term and long-term climate goals before 2030 and 2060 (Dedoussi et al., 2020), respectively. Quantifying the change of regional transport of PM<sub>2.5</sub> under different climate change pathways would be of great interest for a better understanding of the synergy of climate change and air quality, and thus for designing a cooperative control strategy to mitigate both air pollution and climate change.

To fill the gap, this study aims to address two following questions: (1) What is the effect and quantitative variation of acrossregional transport of PM<sub>2.5</sub> among city-clusters across China? 2) How will the regional transport of PM<sub>2.5</sub> respond to the variation of future climate change? Here we conduct multiple simulations to quantify and compare the inter-provincial regional transport of PM<sub>2.5</sub> in China under current (2015) and future (2050) meteorological conditions.

90

#### 2 Data and methods

### 2.1 Model configuration and Methods

The model configurations used in this experiment are as follows. Weather Research and Forecasting Model (WRF, version 4.2) is used to simulate three-dimensional meteorology fields. We follow the same model parameterization scheme as previous study (Liu et al., 2021; Liu et al., 2021), including the Pleim-Xiu land surface physics scheme (Xiu and Pleim, 2001), Asymmetric Convective Mode (Pleim, 2007) planetary boundary layer physics, Morrison double-moment (Morrison et al., 2009) microphysics, Rapid Radiative Transfer Model (Iacono et al., 2008) radiative scheme, and Kain-Fritsch cumulus cloud parameterization (Kain, 2004). A 5-day spin-up simulation is conducted to eliminate the influences of initial condition.

Pollutant concentration distribution is simulated by the latest version of Community Multiscale Air Quality Model (CMAQ, version 5.3.2) configured with the Carbon Bond 6 (CB6) gas-phase chemical mechanism (Sarwar et al., 2008) and AERO6 aerosol module (Sarwar et al., 2008), with improvements in predictions for ozone and other secondary gas-phase species and dry deposition in the latest version (Zheng et al., 2019; EPA, 2020). The Integrated Source Apportionment Method (ISAM) modeling system is used to simulate the distribution, transport, transformation and deposition of aerosols and their precursors. Tagged species in ISAM module are updated by apportioning the change during each chemical and physical process, to track PM<sub>2.5</sub> and its precursors from different emission regions. The ISAM is calculated simultaneously with the CMAQ model (see the

flowchart of ISAM in CMAQ in Fig.1 of Kwok et al (2013)), thus it can well capture the complex interactions among the atmospheric physical and chemical processes during the transport. The source-receptor matrix established with ISAM is able to
reflect the contributions from source to receptor (i.e., the number in transport matrix).

We also estimate the impacts of regional transport at regional level (including several receptors) by taking the weighted average of individual impact, as follows. This formula is intended to be used to calculate the contribution of multi-regional transboundary transport.

$$Contribution = \frac{\sum (transport% \times Concentration_{receptor})}{\sum Concentration_{receptor}},$$
(1)

where *transport* % represents the percentage of regional transport to one receptor, implying the impact of one source to a receptor region; *Concentration*<sub>receptor</sub> represents the baseline concentration of a receptor. So the result denotes the total amount of regional transport received by all receptors in that region.

#### 2.2 Study area

- The modeling domain covers the mainland China with a horizontal spatial grid resolution of 27 km × 27 km (182 × 232 cells) and 14 vertical layers of the meteorological fields which is sufficient to capture the across-region transport at provincial and inter-provincial level (Li et al., 2019). China is divided into 21 quasi-provinces (as some similar administrative provinces have been combined as one single quasi-province, see Supplemental Material Fig. S1a). To analyze regional transport across and within city-clusters, we further combine 21 quasi-provinces into 5 densely populated and severely polluted regions (Fig. S1b): the North China Plain (denoted as NCP, including Beijing-Tianjin-Hebei region, Shandong and Henan province), the Yangtze
- River Delta (denoted as YRD, including Jiangsu-Shanghai region, Zhejiang and Anhui province), the Central China (denoted as HH, including Hubei and Hunan province), the Chengyu area (denoted as CY, including Sichuan province and Chongqing region) and the Feiwen Plain (denoted as FW, including Shanxi and Shaanxi province). The Pearl River Delta (PRD) is not included simply due to its relatively lower PM<sub>2.5</sub> pollution level than other regions (see Fig. 1a).

# 2.3 Data

The FNL (Final) operational global analysis and forecast data from National Centers for Environmental Prediction (NCEP) are used for the 2015 simulation (the baseline scenario) in WRF, the baseline year is the same as our previous study (Liu et al., 2021; Liu et al., 2021). The spatio-temporal resolution of data is on 0.25 degree by 0.25 degree grids prepared operationally every six hours. Assimilated data include global surface and upper observational weather data.

The ensemble averages of five Coupled Model Intercomparison Project (CMIP6) multi-model simulations (i.e., BCC-CSM2-MR

- from China, MRI-ESM2-0 from Japan, IPSL-CM6A-LR and CNRM-CM6-1 from France, and EC-Earth3 from Europe) were used to represent the future meteorological conditions for 2050 in two warming scenarios including SSP126 (low GHG emission pathway) and SSP585 (high GHG emission pathway). We conducted dynamic downscaling to drive the WRF simulations over China domain, following the same configurations and data as our previous study (Liu et al., 2021). Biogenic emission was simulated with the Model of Emissions of Gases and Aerosols from Nature (MEGAN version 2.10)(Liu et al., 2021) driven by
- the WRF simulated meteorological conditions.

 $WRF_{input2050} = fnl_{2015} + \Delta CMIP6_{ssp}$ 

(2)

The monthly anthropogenic emission in this study is the Emission Inventory of Air Benefit and the Cost and Attainment Assessment System (ABaCAS- EI) emission inventory by Tsinghua University for 2015 (Zheng et al., 2019; Xing et al., 2020). We use the same anthropocentric emissions for current and future simulations, so their differences are only considered as the meteorology changes. Four typical months, namely, January, April, July, and October are chosen to represent winter, spring, summer, and autumn, respectively (Li et al., 2019). Here, sensitivity analysis were conducted to fixed on anthropogenic emission, the only change reveal on meteorological conditions (Table 1).

|      |             |          | Table 1 Scenarios                                    |
|------|-------------|----------|------------------------------------------------------|
| Case | Meteorology | Emission | Objective                                            |
| 1    | 2015        | 2015     | Baseline scenario                                    |
| 2    | 2050 SSP126 | 2015     | 2050 friendly climate scenario, current emission     |
| 2    | 2000_001120 | 2015     | 2050 mendry enhance scenario, eurient enhission      |
| 3    | 2050_SSP585 | 2015     | 2050 uncontrolled climate scenario, current emission |

# 135 2.4 Model performance

We compared the CMAQ model simulations with ground observations from the China National Environmental Monitoring Center (http://beijingair.sinaapp.com/) (Supplementary materials Fig. S2). The simulated R-squared (R<sup>2</sup>) equals 0.41, mean fractional bias (MFB) equals -0.39, mean fractional error (MFE) equals 0.41. Although the simulated PM<sub>2.5</sub> concentration tends to be relatively lower compared to the station observations, which is probably due to the uncertainties of secondary aerosol

140

130

formations, the validation results indicate that the WRF-CMAQ model performance is acceptable, as the MFB and MFE are within the recommended standard (MFE $\leq$  +75 % and MFB $\leq$  ±60 %) of Boylan and Russell (2006) and Emery et al (2001) and comparable with previous studies (Chang et al., 2019; Liu et al., 2021; Liu et al., 2021; Dong et al., 2020)

160