# Peer review of "Current and future prediction of inter-provincial transport of ambient PM2.5 in China"

_Atmospheric Chemistry and Physics, 2022_

## Referee Comment (RC2)

Review comments:

This study applied regional CTMs to study the future climate changes on the inter-regional transport of PM2.5 in China. The topic is very interesting. However, the methods described by the authors worried me. In the methods section, the authors listed the equation 2 for the future climate dynamical downscaling. I am not very convinced by the feasibility as described. Are those 5 CMIP6 model outputs were downscaled together and averaged out, or did the authors calculate the climate changes simulated by the 5 CMIP6 models and then add them into the FNL2015 data? If the latter, how is that possible?

When simulating the future climate changes, the authors only ran 4 months (Jan, April, July, October) for the two scenarios, with a few days as spin-up. This is not acceptable to consider the influence of climate variability on the simulation of air pollutants changes.

**Editorial comments:**

L22: "meteorology" to "climate"

L26: change to "suffer"

No Graphical abstract needed for the journal. The short summary is not needed in the manuscript either, but only during submission.

L42: distinguish the three "Wang et al., 2014". Also the paper needs to update the recent studies about the PM2.5 pollution in China.

L46: add "annual mean"

L51: I assume the 411,000 premature deaths was in China?

L70: I am pretty positive that "Dedoussi et al., 2020" study has nothing with China's carbon policy.

L82: distinguish the two Liu et al., 2021 studies. The same as in L116-117.

L88: please find the right reference for the CMAQ AER6 module.

L96: define "regional transport" and "regional level" here. This is very confusing to understand the authors' motivations.

L108-109: The authors regrouped the 21 "quasi-provinces" into five regions, but then the authors claimed that they were studying the city clusters. This is very misleading for the readers.

L129: reorganize the sentence. "their" is not very clear for which was referred here.

L149: "defined as the sum of contribution except for local emission produced" describe how this was calculated.

L150: where are those "densely populated areas"?

L155-156: the explanation for this does not make any sense.

L166: "In general, the largest source of  $PM_{2.5}$  is local contribution," I found this statement is not quite true. If you count the dark colors in Fig 1d, there are 11 out of 21 regions that local emissions dominates more than 50% of total  $PM_{2.5}$ . It seems to me that the local sources are as important as regional transport.

L199: these "inner-regional transport (from nearby provinces within the same region), and across-regional transport" should be defined earlier in the methods. So the readers will understand what the authors are trying to study.

**Table 1:**

Change "2050 friendly climate" to "2050 climate friendly"

Fig. 1. In Fig 1 (d), these abbreviations of provinces in China are hard for the authors to comprehend the message from this plot. The authors probably can add the full names in (b) or (c).

Fig. 6: put all the legends "(a) Source " and "(b) Source" on the same levels.

---

## Author Comment (AC5)

**Response 1**

We thank the reviewer for the valuable and constructive comments and suggestions, which have helped to improve the quality of the manuscript. We have studied comments carefully and made corrections accordingly. Below are our responses to the comments from reviewer. The response follows each comment in black color and questions in blue color.

Reviewer 1:

General comments:

The manuscript entitled "Current and future prediction of inter-provincial transport of ambient PM2.5 in China" presented the current and future states of inter-regional transport (IRT) and across-regional transport (ART) in China. The manuscript title and its content seem to be interesting for many readers at first; however, the modeling configuration which only conducted the fixed emission in 2015 is quite preliminary and will not answer to two questions stated in line 74-77. Without the additional simulation in 2050, the current title, abstract, and introduction will cause misleading. Why were future emissions not used in this study? Because the future emission scenario contains the variability of changes in precursors of PM2.5, I am wondering about the importance of the results drawn from this study. Although this point is discussed in line 325-328 in the conclusion section, this important statement is needed to be carefully introduced. Without the additional simulation in 2050, the title, abstract, and introduction are required to be fully revised. In addition, discussions in future changes are immature at the current presentation quality. The future change of IRT and ART is provided but the reason to cause these changes have not been discussed. What are plausible and/or possible factors to cause these IRT and ART changes in 2050? Please see the following specific points in these general comments.

Response: We appreciate the reviewer's recognition for our work and very useful comments. We have followed all the comments and revised the manuscript accordingly. Please check the following point-by-point responses. The major concern of the reviewer is about the simulation in 2050 with variation of emissions. We have followed the reviewer's suggestion and conducted the CMAQ-ISAM simulation in 2050 with the new emission scenario (co-benefit energy scenario, CBE). In addition, as the reviewer suggested, we have provided more discussion about the possible factors to cause the future changes in IRT and ART. The results of the additional simulation and discussion have been added to the revised manuscript, and also presented in the details to the following specific comments.

Specific comments:

Line 82: In my understanding, input data for Pleim-Xiu LSM is limited in the application over the U.S. How did the authors prepare for the necessary input data?

Response: Thanks for the suggestion. The Pleim-Xiu LSM is applicable to the global scale and its performance has been well examined in our previous simulations over

the northern hemisphere (Xing et al., 2015) and the China domain (Liu et al., 2021a).

Reference

Xing, J., Mathur, R., Pleim, J., Hogrefe, C., Gan, C.-M., Wong, D. C., Wei, C., Gilliam, R., and Pouliot, G.: Observations and modeling of air quality trends over 1990–2010 across the Northern Hemisphere: China, the United States and Europe, Atmos. Chem. Phys., 15, 2723–2747, https://doi.org/10.5194/acp-15-2723-2015, 2015.

Liu, S., Xing, J., Wang, S., Ding, D., Cui, Y., and Hao, J.: Health Benefits of Emission Reduction under 1.5 degrees C Pathways Far Outweigh Climate-Related Variations in China, Environ. Sci. Technol., 55, 10957-10966, 10.1021/acs.est.1c01583, 2021a.

Line 85: But what kind of initial condition is used in this study? Please state. In addition, I guess that the selection of boundary conditions is also an important aspect of this simulation. How did the authors prepare the lateral boundary condition? Are the contributions from the lateral boundary condition (i.e., outside of China) small? If there are some contributions, I am also wondering how can we consider the global change of PM2.5 in future status.

Response: We use the CMAQ clean profiles for both initial (ICON) and boundary conditions (BCON). We spin up 7 days to eliminate the influence of the ICON. There are indeed some influences from outside of China but much less foreign impacts are found in the five target regions as we presented in our previous studies using hemispheric CMAQ simulations (Liu et al, 2020). Since this study mainly focused on the inner-provincial transport rather than the cross-country study, also it is very difficult to make accurate prediction as the emissions scenarios are hard to predict for other countries, here we used the fixed clean boundary condition profile for all the simulations in this study. But it might be another very interesting study for the climate-driven cross-country transport of $PM_{2.5}$ in future analysis.

We have clarified this point in the revised manuscript as follows.

(Page 4, Line 87-90) "A 7-day spin-up simulation is conducted to eliminate the influences of initial condition. We use the clean profiles for lateral boundary condition in all simulations, as the foreign impacts (outside China) are very small in the five target regions (Liu et al, 2020). Though we didn't consider the influence of foreign impacts on China in this study, further studies are suggested to investigate the climate-driven cross-country transport of $PM_{2.5}$ in future."

Reference

Liu, S., Xing, J., Wang, S., Ding, D., Chen, L., & Hao, J. (2020). Revealing the impacts of transboundary pollution on PM2. 5-related deaths in China. Environment international, 134, 105323.

Line 94-95: It should be clearly introduced the limitation of ISAM method to estimate contributions. For example, source sensitivities which are critically important for secondary aerosols were not evaluated by ISAM. Please add the details of the

Response: Thanks for the reviewer's suggestion. In the revised manuscript, we clarify that there is an underestimation in the CMAQ simulation (section 2.4), probably due to the uncertainty in the formation of secondary aerosols, and of course, the flaws in the CAMQ model are also reflected in the ISAM model.

In addition, we also demonstrated the advantages of ISAM in section 2.1 (line 100-108): Unlike the zero-out method must be run multiple times in different scenarios, which is expensive and time-consuming to compute, the major advantages of ISAM method can accurately identify the contribution of each type of source or each region, without the need for multiple calculations, making the calculation more efficient. For near-linear systems (EC, OC, $SO_2$, $NH_3$, NOx), ISAM compares well with the zero-out method, but ISAM compares less well for nonlinear systems (ozone, sulfate, nitrate and ammonium). The major disadvantage of ISAM model is the secondary aerosols. Furthermore, indirect effects from interactions between various inorganic particulate matter components, which cause the non-linearity and are not included in ISAM by design (Kwok et al., 2013). Overall, however, the ISAM method is suitable for efficient calculations to assess the impact of different emission sources or different areas on the atmospheric environment.

Reference

Kwok R H F, Napelenok S L, Baker K R. Implementation and evaluation of PM2. 5 source contribution analysis in a photochemical model[J]. Atmospheric Environment, 2013, 80: 398-407.

As the reviewer suggested, we have clarified this limitation in the revised manuscript as follows.

(Page 4, Line 100-108) "Unlike the zero-out method must be run multiple times in different scenarios, which is expensive and time-consuming to compute, the major advantages of ISAM method can accurately identify the contribution of each type of source or each region, without the need for multiple calculations, making the calculation more efficient. For near-linear systems (EC, OC, $SO_2$, $NH_3$, NOx), ISAM compares well with the zero-out method, but ISAM compares less well for nonlinear systems (ozone, sulfate, nitrate and ammonium).The major disadvantage of ISAM model is about the secondary aerosols were not evaluated by ISAM. Furthermore, indirect effects from interactions between various inorganic particulate matter components, which cause the non-linearity and are not included in ISAM by design (Kwok et al., 2013). Overall, however, the ISAM method is suitable for efficient calculations to assess the impact of different emission sources or different areas on the atmospheric environment."

Line 99: The term "transport" cannot be followed. How was derived? Please explicitly explain and define.

Response: We thank the reviewer for this valuable comment. We changed this variable name from "transport" to "Ma" in following equation (Page 6).

$$Multi\_contribution = \frac{\sum(M_a \% \times C_k)}{\sum C_k} \tag{5}$$

Here, $M_a$ % is the value from Figure 1d. $C_k$ represents the baseline CMAQ concentration of a receptor. So the equation (5) denotes the total amount of regional transport and interaction in five targets regions (see Fig. S1b). Results are shown in Fig. 3 and Fig.7.

Line 105: Need the height depth of the lowest layer.
Response: The height depth of the lowest layer is about 38m.
We have clarified this point in the revised manuscript as follows.
    (Page 4, Line 111) "14 vertical layers (the lowest layer is about 38 m)"

Line 127: The biomass burning sources are not taken in this modeling system? If yes, why?
Response: Biomass burning sources are taken in our modeling system, which belongs to anthropogenic emission inventory. According to a bottom-up approach, our ABaCAS emission inventory classification included 16 sectors (fertilizer application, livestock, domestic (bio-fuel), domestic (fossil fuel), domestic solvent use, other domestic use, industry combustion, open burning, power plant, cement, steel, other industry process, industry solvent use, other stack industry process, off-road transport and on-road transport).

    For the sector of domestic biomass burning, we established a dataset based on three independent nationwide statistics/surveys, as described in our previous study(Zheng et al., 2019b; Zhao et al., 2018; Xing et al., 2020)."

    Given the importance of biomass burning, the main sources of $PM_{10}$ and $PM_{2.5}$ emissions were industrial combustion and domestic combustion in 2005, with higher contributions from domestic combustion in biomass burning. After 2013, the consumption of domestic (bio-fuel) decreased significantly, leading to a reduction in particulate matter emissions from the domestic combustion sector.

We have clarified this point in the revised manuscript as follows.
    (Page 5, Line 143-146) "including 16 sectors (fertilizer application, livestock, domestic bio-fuel combustion, domestic fossil fuel combustion, domestic solvent use, other domestic use, industry combustion, open burning, power plant, cement, steel, other industry process, industry solvent use, other stack industry process, off-road transport and on-road transport). (Zheng et al., 2019b; Zhao et al., 2018; Xing et al., 2020)."

Reference

Zhao B, Zheng H, Wang S, et al. Change in household fuels dominates the decrease in PM2. 5 exposure and premature mortality in China in 2005－2015[J]. Proceedings of the National Academy of Sciences, 2018, 115(49): 12401-12406.

Zheng H, Cai S, Wang S, et al. Development of a unit-based industrial emission inventory in the Beijing–Tianjin–Hebei region and resulting improvement in air quality modeling[J]. Atmospheric Chemistry and Physics, 2019, 19(6): 3447-3462.

Xing J, Lu X, Wang S, et al. The quest for improved air quality may push China to continue its CO2 reduction beyond the Paris Commitment[J].Proceedings of the National Academy of Sciences of the United States of America,2020,117(47): 29535-29542.

Line 129-130: The configuration of emission in this study is preliminary because the simulation was only conducted without changing emissions in 2050. In this sense, the title of the manuscript will mislead, and the abstract and introduction section are needed to be carefully written to state that the future change is only derived from the meteorological condition.

Response: We thank the reviewer for this valuable suggestion. We agree with the reviewer that the original design without considering the change in future emissions is quite preliminary. As the reviewer suggested, we have conducted additional simulations (the last three cases) with the future emission scenarios and revised all the results in the revised manuscript. The updated table1 is as follows.

Table 1  Summary of scenarios designed in this study

| Case | Meteorology | Emission | Objective |
| --- | --- | --- | --- |
| 2015-Base | 2015 | 2015 | Baseline scenario |
| 2050-SSP126-REF | 2050_SSP126 | REF (same as 2015) | 2050 climate friendly scenario, current pollutant emission |
| 2050-SSP585-REF | 2050_SSP585 | REF (same as 2015) | 2050 uncontrolled climate scenario, current pollutant emission |
| 2015-CBE | 2015 | CBE | controlled pollutant emission scenario |
| 2050-SSP126-CBE | 2050_SSP126 | CBE | 2050 climate friendly scenario, controlled pollutant emission |
| 2050-SSP585-CBE | 2050_SSP585 | CBE | 2050 uncontrolled climate scenario, controlled pollutant emission |

Through multi-scenario future simulations, we found some key conclusion as follows.

(1) when we consider both the roles of emission reductions and climate friendly in the future, 2050-SSP126-CBE plays the largest role in the co-benefits of reducing the magnitude of $PM_{2.5}$ regional transport.

We found that the 2050-SSP585-REF scenario is similar to the current 2015 baseline while 2050-SSP126-CBE shows significant reductions in regional transport. Specifically, the average transported PM$_{2.5}$ concentration in 2050-SSP585-REF is about 7.45 µg/m$^3$ which is solely driven by the effect of meteorological fluctuations (Fig 4a). However, the PM$_{2.5}$ regional transport significant drops to an average concentration of 2.49 µg/m$^3$ (Fig 4b), highlighting the co-benefits of 2050-SSP126-CBE in reducing the magnitude of PM$_{2.5}$ regional transport.

[Figure]

**Figure 4.** The PM$_{2.5}$ regional concentration (a,b,c) and matrix (d,e,f) under 2050-SSP585-REF, 2050-SSP126-CBE and their differences.

(2) emission reductions lead to a substantial reduction in PM$_{2.5}$ regional transport, which far outweigh the influences of meteorological fluctuations driven by future climate change (Fig 5).

Specifically, controlling pollutant emission is undoubtedly reduced the concentration of PM$_{2.5}$ regional transport, in particular, the maximum reduction occurs in the NCP region (15.99 µg/m$^3$), followed by YRD (15.02 µg/m$^3$), HH (14.75 µg/m$^3$), CY (12.47 µg/m$^3$) and FW (8.08 µg/m$^3$). However, the effects of meteorological fluctuations are still uncertain range (-7.19 µg/m$^3$ to 2.59 µg/m$^3$). Therefore, results suggest that the changes in regional transport from emission controls are much greater than that due to future meteorological fluctuations.

[Figure]

**Figure 5** The average and variation range of Δemission-driven (a,b) and Δmeteorology-driven (c,d) impacts on the change of PM$_{2.5}$ regional transport.

We have clarified this point in the revised manuscript as follows.

(Page5, line147-154) "Future emissions under controlled pathway is the same as our previous study which estimates the co-benefits of energy policy in reducing air pollution (noted as co-benefit energy scenario, CBE).Specifically, in comparison with the REF scenario, the CBE decreases the PM$_{2.5}$ (73.3%), SO$_2$ (77.6%), NOx (77.3%), and VOC emission (60.0%) in 2050. Here, sensitivity analysis was conducted with different combination of meteorology and emission scenarios (Table 1). The same baseline anthropocentric emissions are used in Case-Base, Case-2050-SSP126-REF and Case-2050-SSP585-REF but with different meteorological conditions, indicating their differences are only driven by the meteorology changes. Similarly, only meteorology varies in Case-2015-CBE, Case-2050-SSP126-CBE and Case-2050-SSP585-CBE which are used to analyze the change in emissions (Xing et al., 2020; Liu et al., 2021a) We simulated 12 months as a year."

In addition, we have clarified this point in the revised abstract as follows.

(Page2 line31-36) "Controlling pollutant emission is undoubtedly reduced the concentration of PM$_{2.5}$ regional transport (by -5.25 μg/m$^3$ on average), largely exceeding the influence from the meteorological fluctuations (by -0.82 μg/m$^3$ on average) driven by the climate change in 2050s. On the other hand, future controlled pollutant emissions proved that strengthen across-regional transport with an enlarged relative contribution to total PM$_{2.5}$ concentration (8.46%) in 2050, along with a

decreased contribution from local sources (8.54%).""

Response: Thanks for reviewer's recommendation. We have added this reference and contents in revised manuscript as follows.

(Page 6, Line 173-175) "According to Huang (2021), root mean square error (RMSE, 118 studies), MFB and MFE are frequently used (>20 studies) metrics. And over half of the articles used all three statistical metrics for model performance evaluation."

Response: Thanks for the suggestion. In order to accurately describe the seasonal variation, we revised the sentence as follows.

(Page 7, Line 185-186) "Seasonality was also observed, with the worst severe pollution in winter, slight differences were shown in fall, spring, and summer (Fig. S3)."

Response: We thank the reviewer for pointing out the confused definitions, the "contribution" here are indeed different from the definition in Eq.(3). To avoid confusion, we have defined the "contribution" in the revised manuscript as follows.

(Page 6)

$$R_{k(i.j)} = \sum_{1}^{21} I_{m(i,j)} - I_{k(i,j)} \tag{3}$$

$$SA_{k(i,j)} = \frac{R_{k(i,j)}}{S_{k(i,j)}} \times 100\% \tag{4}$$

where $R_{k(i,j)}$ represents the sum of non-local $PM_{2.5}$ concentrations, $I_{k(i,j)}$ and $I_{m(i,j)}$ are the gridded ISAM model concentration with tagged local and sum of the 21 regions shown in Figure S1a, $S_{k(i,j)}$ represents the simulated total $PM_{2.5}$ concentration from the ISAM model So equation (3) result denotes the total concentration ($\mu g/m^3$) of regional transport received by all receptors in that region. Equation (4) denotes the percentage of $PM_{2.5}$ regional transport in all other regions. Unlike the previous

calculation, which used the concentration of the CMAQ model. Besides, we simulate all 12 months and use new initial conditions of the CMAQ-ISAM model. Therefore, transport contribution is higher than the previous results (Fig 1c). We changed the original equation (2) of the manuscript to equation (1).

Line 236-257 (Section 3.3): I cannot follow the reason for increasing PM2.5 in 2050. Does only the weakened atmospheric circulation attribute to this increase? Is there no impact from other meteorological factors such as temperature, relative humidity, precipitation, and wind field? The discussion here is just the provision of model simulation results and is quite immature to understand results.

Response: We thank the reviewer for this valuable comment. Our original simulation was only focused on four months and one ensemble run for each scenario which might not be enough for eliminating the uncertainties in future prediction analysis. In the revised manuscript, we conducted multiple simulations with five different climate model predictions, and run through the whole year (according to review2's suggestion). We conducted a series of experiments as shown in Table 1, the difference between three emission conditions (i.e., 2015-CBE vs 2015-Base; 2050-SSP126-CBE vs 2050-SSP126-REF; 2050-SSP585-CBE vs 2050-SSP585-REF), and the differences between five multi-climate models (BCC, MRI, IPSL, EC, CNRM) under high emissions (even smaller influences of are expected under low emission, see Fig. S8).

We compared the $PM_{2.5}$ total changes due to the meteorology in the future varies significantly in future under different simulations, as Figure S5 in the supplemental material. Through the multi-model ensemble simulation, we find that future $PM_{2.5}$ changes are variable. For example, a significant dipole distribution exists in the IPSL model, the EC model also displays slight dipole distribution, MRI and CNRM overall show a decreasing $PM_{2.5}$ pattern.

Therefore, future climate-driven changes in $PM_{2.5}$ concentrations are complex. Many meteorological factors influence the distribution of $PM_{2.5}$, such as precipitation and relative humidity as the reviewer's comment, these factors have fluctuation ranges, moreover, synergy or offset effects also exist in meteorological factors (Liu et al., 2021a). In general, precipitation and planetary boundary layer can reduce the $PM_{2.5}$ concentration by wet deposition and convection, respectively. On the contrary, the increase in temperature brings about an enhanced atmospheric oxidation and kinetics reaction, thus potentially leading to an increase in secondary $PM_{2.5}$ concentrations. Furthermore, lower relative humidity results in weaker hygroscopic growth and may increase the $PM_{2.5}$ concentration in the future.

Here we show the difference in the ensemble mean of precipitation (mm), planetary boundary layer height (m) and relative humidity (%) between the 2015 baseline and 2050 projected (Figure S10), temperature is shown in Figure S10, all these changes present a similar pattern to our previous study (Liu et al., 2021a). Considering the variability of meteorological conditions, the future of $PM_{2.5}$ will change as well.

The future slight change of $PM_{2.5}$ concentration is associated with the higher planetary boundary layer height and increased precipitation in the 2050 global warming trend

(Fig. S10), these changes are consistent with IPCC's recent reports and with high confidence (Thackeray et al., 2022), highlighting that global warming intensifies the occurrence of extreme precipitation in the future, particularly in China (Takahashi et al., 2020; Qin et al., 2021).

Reference

Liu S, Xing J, Wang S, et al. Health benefits of emission reduction under 1.5° C pathways far outweigh climate-related variations in China [J]. Environmental Science & Technology, 2021a, 55(16): 10957-10966.

Qin, P., Xie, Z., Zou, J., Liu, S., and Chen, S.: Future Precipitation Extremes in China under Climate Change and Their Physical Quantification Based on a Regional Climate Model and CMIP5 Model Simulations, Adv. Atmos. Sci., 38, 460-479, 10.1007/s00376-020-0141-4, 2021.

Takahashi, H. G., Kamizawa, N., Nasuno, T., Yamada, Y., Kodama, C., Sugimoto, S., and Satoh, M.: Response of the Asian Summer Monsoon Precipitation to Global Warming in a High-Resolution Global Nonhydrostatic Model, J. Climate, 33, 8147-8164, 10.1175/JCLI-D-19-0824.1, 2020.

Thackeray, C. W., Hall, A., Norris, J., and Chen, D.: Constraining the increased frequency of global precipitation extremes under warming, Nat. Clim. Change, 12, 441-448, 10.1038/s41558-022-01329-1, 2022.

total PM$_{2.5}$

[Figure]

**Figure S5.**    The distribution of annual average differences in simulated PM$_{2.5}$ total concentrations between the SSP126 predicted scenarios (BCC, MRI, IPSL, EC, CNRM models) and the SSP585 pathways.

[Figure]

**Figure S10.** The difference in precipitation (mm), planetary boundary layer height (m)   and relative humidity (%) between 2050 projected and 2015 baseline.

The right panel is the difference between SSP126 and 2015, the middle is the difference between SSP585 and 2015, and the right panel is the difference between SSP126 and SSP585 scenarios.

(a), (b) and (c) show the variation of precipitation (mm); (d), (e) and (f) show the variation of planetary boundary layer height (m); (g), (h) and (i) show the variation of relative humidity (%).

Therefore, to clarify this point, we have re-write all section 3.2 in the revised manuscript.

Line 262-310 (Section 3.4): Similar to the above comment, I cannot understand what

causes these future changes in regional PM2.5 transport. Of course, it is hard to seek the explicit reason to lead these changes, but the current discussion lacks plausible and/or possible reasons for future changes.

Response: We thank the reviewer for the good comment. According to the new experimental design, we reorganized the original section 3.4 into an updated section 3.2.2, and the updated figures are Figures 6 and 7, and added a new Figure 5 and Figure S5 and S6.

We compared the $PM_{2.5}$ regional transport due to the meteorology in future varies significantly in the future under different simulations (Figure S6). Same as Figure S5, different $PM_{2.5}$ spatial distributions are presented in different climate models, for example, a significant dipole distribution exists in the IPSL model, and other models such as CNRM and EC present significantly decreasing patterns in the future.

Here we illustrate the basic distribution of regional transport. The explanation of IRT and ART changes in 2050 are discussed on page 16-18.

First of all, emissions reductions play a dominant role in weakening regional transport, resulting in the decreasing local contribution and increasing across-regional transport (ART). However, the impacts of future climate change are complex and variable. Moreover, to reduce the impact of future uncertainty, we have added a fluctuation range for each factor. Apparently, the range of climate fluctuation is larger than emission roles.

$PM_{2.5}$ regional transport is mainly driven by monsoon circulation and pollutant concentration from upwind regions. Considering the differences between seasons, we calculate the seasonal mean of the surface wind field (Figure S11), including the difference in atmospheric circulation between 2050 projected and the differences between the 2015 baseline and 2050 projected. It is notable that enhanced southerly wind, weakens the east Asia winter monsoon (EAWM) in both SSP126 and SSP585 scenarios (Figure S11 a, b). However, the east Asia summer monsoon (EASM) display is slightly enhanced (Figure S11g, h). Our results are consistent with Wu (2020), who indicated that the trends of wind speed averaged over China will decrease significantly in both annual and winter under all three scenarios in the middle and late 21 st century, but it will increase significantly in summer under SSP585.

Furthermore, we suggest that a weakened wind field in the context of global warming may be related to a weakening of temperature gradients between the equator and the poles, leading to a weakening of circulation. Previous studies demonstrated that global warming results in a weakening of the global atmospheric circulation such as Hadley Cell (Kim et al., 2022; Wang, 2004; Levine and Schneider, 2011; Kim et al., 2020; Hu et al., 2018).

Reference

Hu, Y., Huang, H., and Zhou, C.: Widening and weakening of the Hadley circulation under global warming, Science Bulletin, 63, 640-644, https://doi.org/10.1016/j.scib.2018.04.020, 2018.

Kim, D., Kim, H., Kang, S. M., Stuecker, M. F., and Merlis, T. M.: Weak Hadley cell intensity changes due to compensating effects of tropical and extratropical radiative forcing, NPJ CLIMATE AND ATMOSPHERIC SCIENCE, 5, 10.1038/s41612-022-00287-x, 2022.

Kim, H., Ha, K., Moon, S., Oh, H., and Sharma, S.: Impact of the Indo-Pacific Warm Pool on the Hadley, Walker, and Monsoon Circulations, Atmosphere-Basel, 11, 10.3390/atmos11101030, 2020.

Levine, X. J., and Schneider, T.: Response of the Hadley Circulation to Climate Change in an Aquaplanet GCM Coupled to a Simple Representation of Ocean Heat Transport, J. Atmos. Sci., 68, 769-783, 10.1175/2010JAS3553.1, 2011.

Wang, C. Z.: ENSO, Atlantic climate variability, and the Walker and Hadley circulations, HADLEY CIRCULATION: PRESENT, PAST AND FUTURE, 21, 173-202, 2004.

Wu, J., Shi, Y., and Xu, Y.: Evaluation and Projection of Surface Wind Speed Over China Based on CMIP6 GCMs, JOURNAL OF GEOPHYSICAL RESEARCH-ATMOSPHERES, 125, 10.1029/2020JD033611, 2020.

regional transport PM$_{2.5}$

[Figure]

Figure S6 The left panel shows the distribution of annual average differences in simulated PM$_{2.5}$ regional transport concentrations between the SSP126 predicted scenarios (BCC, MRI, IPSL, EC, CNRM models) and the SSP585 pathways. The right panel shows the regional transport of PM$_{2.5}$ concentrations simulated by CMIP6 multi-climate models (μg/m3), including NCP, YRD, HH , CY and FW five target regions.

[Figure]

**Figure S11.**    Similar to Figure S10, the near-surface wind field and temperature simulation in various between the 2015 baseline and 2050 projected. Temperature is represented by shading color (°C).

(a), (b) and (c) show winter (December, January, February, DJF); (d), (e) and (f) show spring (March, April, May, MAM); (g), (h) and (i) show summer(June, July, August, JJA); (j), (k) and (l) show autumn (September, October, November, SON), respectively.

 I agree that the emission reduction can further weaken regional transport, but the reduction rate highly depends on provinces/regions. It might not be simple to state so, therefore, I think the explicit simulation in 2050 is necessary within this study.

Response: We thank the reviewer for the good suggestion. Generally, we think that emission reduction will attenuate $PM_{2.5}$ regional transport, but it is also just a conjecture, to confirm this point, we agree with the reviewer that it is necessary to add the supplement experiment of 2050, the future low emission scenario (co-benefit energy scenario, CBE scenario), which refers to Liu et al (2021a) and Xing et al (2020). The updated table1 is as follows.

Table 1    Summary of scenarios designed in this study

| Case | Meteorology | Emission | Objective |
| --- | --- | --- | --- |
| 2015-Base | 2015 | 2015 | Baseline scenario |
| 2050-SSP126-REF | 2050_SSP126 | REF (same as 2015) | 2050 climate friendly scenario, current pollutant emission |
| 2050-SSP585-REF | 2050_SSP585 | REF (same as 2015) | 2050 uncontrolled climate scenario, current pollutant emission |
| 2015-CBE | 2015 | CBE | controlled pollutant emission scenario |
| 2050-SSP126-CBE | 2050_SSP126 | CBE | 2050 climate friendly scenario, controlled pollutant emission |
| 2050-SSP585-CBE | 2050_SSP585 | CBE | 2050 uncontrolled climate scenario, controlled pollutant emission |

Our supplement scenarios proved that pollutant emissions reduction will attenuate the absolute $PM_{2.5}$ regional transport, emission reductions lead to a substantial reduction in $PM_{2.5}$ absolute concentration, which far outweighs the influences of meteorological fluctuations driven by the future climate change. Furthermore, the declined $PM_{2.5}$ concentration caused by emission reductions dominates the substantial reduction in $PM_{2.5}$ regional transport. However, we found that the percentage of regional transport would strengthen in the future, resulting in environmental inequality issues.

To address the reviewer's concern, we noted that slight future increase in $PM_{2.5}$ ART concentrations in the HH and CY regions under climate change factors (Figure 6). Combined with the changes in the wind field we found that the increase in $PM_{2.5}$ in these regions is attributed to the enhanced northerly wind (Figure S11i) bringing more northern pollution (generally considered more serious in northern areas such as BTH regions). The wind direction in other seasons does not show a clear pattern. For inner-regional transport (IRT), take HH regions as an example, with the strengthening of the north wind, the pollution transport from Hubei (source) to Hunan (receptor) is increased, while the transport from Hunan(source) to Hubei (receptor) is weakened.

We have revised the manuscript accordingly.

(Page14-15, line 324-333) "Fig 6 summarizes regional transports and interactions of PM$_{2.5}$ in five key regions in 2050. Similarly, the $\Delta$emission-driven impacts are much larger than the $\Delta$meteorology-driven impacts on regional transport in all five regions. The local contributions (noted as $\Delta$Local in Fig 6) are substantially reduced (-4.5% to -10.4%) due to the emission reduction, which far outweighs the influence of meteorological fluctuations (-2.9% to 2.5%). Correspondingly, across-regional transport contributions (noted as $\Delta$ART in Fig 6) are increased (3.3% to 11.8%) by the emission reduction in all five regions. However, the inner-regional transport contributions (noted as $\Delta$IRT in Fig 6) are barely changed (-2.5% to 2.3%) due to either emission reduction or meteorological fluctuations. Apparently, along with the future emission controls, the reduction in local contributions will also lead to an enhancement in the contribution from across-regional transport rather than the inner-regional transport. Therefore, across-regional transport (ART) contributions rise significantly under future strict control of pollutant emissions, such results highlighted the significance of joint-provincial cooperation."

[Figure]

**Figure 6.** Comparison of Δemission-driven and Δmeteorology-driven impacts on local contributions (ΔLocal), Inner-regional Transport (ΔIRT) and Across-regional Transport (ΔART) in five key regions.

We have clarified this point in the revised manuscript as follows.

(Page 16-17, line343-352) "The future change of the interactions among key regions is shown in Fig. 7. Similarly, emission controls significantly reduce the across regional interactions, which largely outweighs the influence of the meteorological fluctuations. Among all five regions, the NCP presents the greatest change due to the Δemission-driven (Fig. 7a and 7b). More specifically, the absolute decreases the most by 12.09 μg/m³, followed by NCP to HH receptor (8.13 μg/m³). However, the relative impacts is increased in most of regions, as the impacts of NCP to YRD increased by 2.69 %, and the NCP to HH receptor increased by 2.67 %. There were fewer changes due to Δmeteorology-driven, with a slight increase of 0.78 μg/m³ (0.29 %) from NCP

to YRD receptor and 0.64 µg/m³ (0.73 %) from NCP to HH receptor, which is much smaller compared to the influence of Δemission-driven. The CY region shows the weakest connection with other areas, with the least variation (less than 1 %) in 2050 scenarios. Less PM₂.₅ across-regional transport to CY and FW regions in future emission and climate change, implying the decreased PM₂.₅ concentrations from other sources to CY and FW regions."

[Figure]

**Figure 7.** Similar to Figure 3, the relationship between five sources and receptors in 2050 (red and yellow bar show Δemission-driven, blue and green bar show Δmeteorology-driven). Each subplot represents the effect of a single source on the other four receptors.

Reference:

Liu S, Xing J, Wang S, et al. Health benefits of emission reduction under 1.5° C pathways far outweigh climate-related variations in China[J]. Environmental Science & Technology, 2021a, 55(16): 10957-10966.

Xing J, Lu X, Wang S, et al. The quest for improved air quality may push China to continue its CO2 reduction beyond the Paris Commitment[J].Proceedings of the National Academy of Sciences of the United States of America,2020,117(47): 29535-29542.

Technical corrections:
Line 82 and elsewhere in this manuscript: Liu et al. (2021) should be distinguished.
Response: We thank the reviewer for pointing out the confused reference.

Liu, S., Xing, J., Wang, S., Ding, D., Cui, Y., and Hao, J.: Health Benefits of Emission Reduction under 1.5 degrees C Pathways Far Outweigh Climate-Related Variations in China, Environ. Sci. Technol., 55, 10957-10966, 10.1021/acs.est.1c01583, 2021a.

Liu, S., Xing, J., Westervelt, D. M., Liu, S., Ding, D., Fiore, A. M., Kinney, P. L., Zhang, Y., He, M. Z., Zhang, H., Sahu, S. K., Zhang, F., Zhao, B., and Wang, S.: Role of emission controls in reducing the 2050 climate change penalty for PM2.5 in China, Sci. Total Environ., 765, 10.1016/j.scitotenv.2020.144338, 2021b.